# Critical behaviour of the stochastic Wilson-Cowan model

**Antonio de Candia**[1,2]*, **Alessandro Sarracino**[3], **Ilenia Apicella**[1,2], **Lucilla de Arcangelis**[3]

**1** Dipartimento di Fisica "E. Pancini", Università di Napoli Federico II, Napoli, Italy, **2** Istituto Nazionale di Fisica Nucleare (INFN), Sezione di Napoli, Gruppo collegato di Salerno, Fisciano, Italy, **3** Dipartimento di Ingegneria, Università della Campania "Luigi Vanvitelli", Aversa, Italy

\* decandia@na.infn.it

**Data Availability Statement:** All relevant data are within the manuscript. The numerical codes for the Gillespie algorithm and the correlation function can be found at http://people.na.infn.it/~decandia/codice/cowan/ploscompbio/code.zip.

## Abstract

Spontaneous brain activity is characterized by bursts and avalanche-like dynamics, with scale-free features typical of critical behaviour. The stochastic version of the celebrated Wilson-Cowan model has been widely studied as a system of spiking neurons reproducing non-trivial features of the neural activity, from avalanche dynamics to oscillatory behaviours. However, to what extent such phenomena are related to the presence of a genuine critical point remains elusive. Here we address this central issue, providing analytical results in the linear approximation and extensive numerical analysis. In particular, we present results supporting the existence of a bona fide critical point, where a second-order-like phase transition occurs, characterized by scale-free avalanche dynamics, scaling with the system size and a diverging relaxation time-scale. Moreover, our study shows that the observed critical behaviour falls within the universality class of the mean-field branching process, where the exponents of the avalanche size and duration distributions are, respectively, 3/2 and 2. We also provide an accurate analysis of the system behaviour as a function of the total number of neurons, focusing on the time correlation functions of the firing rate in a wide range of the parameter space.

## Author summary

Networks of spiking neurons are introduced to describe some features of the brain activity, which are characterized by burst events (avalanches) with power-law distributions of size and duration. The observation of this kind of noisy behaviour in a wide variety of real systems led to the hypothesis that neuronal networks work in the proximity of a critical point. This hypothesis is at the core of an intense debate. At variance with previous claims, here we show that a stochastic version of the Wilson-Cowan model presents a phenomenology in agreement with the existence of a bona fide critical point for a particular choice of the relative synaptic weight between excitatory and inhibitory neurons. The system behaviour at this point shows all features typical of criticality, such as diverging timescales, scaling with the system size and scale-free distributions of avalanche sizes and durations,

**Funding:** AdC, IA and LdA acknowledge financial support from the MIUR PRIN 2017WZFTZP. AS acknowledges financial support form MIUR PRIN PRIN201798CZLJ. The funders had no role in study design, data collection and analysis, decision to publish, or preparation of the manuscript.

**Competing interests:** The authors have declared that no competing interests exist.

with exponents corresponding to the mean-field branching process. Our analysis unveils the critical nature of the observed behaviours.

## Introduction

Spontaneous brain activity shows complex spatio-temporal patterns characterized by a rich phenomenology, including power-law spectra [1], instabilities and metastability transitions [2–4], synchronization [5, 6], the presence of multiple spatio-temporal scales [7, 8], etc. Another striking feature is the occurrence of bursts, or avalanches, as first observed in organo-typic cultures from coronal slices of rat cortex [9]. This kind of behaviour has been confirmed in a wide variety of systems, from cortical activity of awake monkeys [10] to human fMRI (functional Magnetic Resonance Imaging) [11] and MEG (MagnetoEncephaloGraphy) recordings [12]. In experiments, the distribution of avalanche size $S$ is characterized by the scaling law $P(S) \sim S^{-\tau_S}$ with exponent $\tau_S \simeq 1.5$, whereas the distribution of the avalanche duration $T$ follows the scaling $P(T) \sim T^{-\tau_T}$ with $\tau_T \simeq 2$. Both these behaviours are consistent with the universality class of the mean-field branching process [13], where the propagation of an avalanche can be described by a front of independent sites that can either trigger further activity or die out.

The hypothesis that some features of the brain activity can be interpreted as the result of a dynamics acting close to a critical point has inspired several statistical models where a critical state can be selected by the fine tuning of a parameter [14–16], or is self-organized [17–20]. Numerical data for different neuronal network models well reproduce experimental results. On the other hand, other stochastic models have been proposed that can reproduce the avalanche dynamics of the neural activity, without invoking the existence of an underlying critical behaviour. Among these models of spiking neurons, a central role is played by the celebrated Wilson-Cowan model (WCM), which describes the coupled dynamics of populations of excitatory and inhibitory neurons [21–24]. One of the major merits of this model is that it allows for analytical treatment in the large population size limit [22, 25, 26]. The stochastic version of this model has been shown to reproduce avalanche dynamics [26] and oscillatory behaviour of the activity [27]. However, the underlying mechanisms responsible for such phenomenology have been identified in the noisy functionally coupled structure of the dynamics, rather than in the presence of a critical point.

### Requirements to assess critical behavior

In order to clarify the main requirements for the behaviour of a system to be classified as critical, here we briefly summarize the fundamental features of criticality [28]. Second order (critical) phase transitions are characterized by singularities in the proximity of a specific value of a control parameter, for instance the temperature in thermal phase transitions, in the limit of infinite system sizes and, generally, for vanishing external fields. More explicitly, fundamental properties of the system either diverge or go to zero approaching the critical value of the parameter. In particular, the order parameter of the system goes continuously to zero at the critical point, being non-zero only on one side of the transition (low temperatures in thermal systems), whereas the response function, proportional to the fluctuations of the order parameter, diverges. Fundamental requirement of second order phase transitions is that this singular behaviour is described by a function that in the neighbourhood of the critical point can be approximated by a power law, neglecting terms of higher order representing corrections to scaling. This property allows one to define a critical exponent for each quantity of interest and

therefore a family of critical exponents characterizing the critical behaviour, named universality class. Different systems can belong to the same universality class if they are described by Hamiltonians with the same symmetries. Since critical transitions occur in systems of interacting components, the divergence of the response function implies, by fluctuation-dissipation relations, that at the critical point the spatial and temporal correlation ranges diverge in an infinite system. The divergence of the temporal correlation range is expression of the well-known critical slowing down taking place at the critical point, whereas the divergence of the correlation length expresses the large scale sensitivity of the system to external perturbations.

In finite systems, the spatial correlation range at criticality equals the system size. As a consequence, a diverging (or equal to the system size) correlation length implies that no characteristic size exists in the system and therefore the extension of the power law regime, namely the cutoff, must scale with the system size. Therefore, the divergence of the correlation length and the absence of a characteristic size are reflected in the power law behaviour of characteristic distributions. Summarizing, to assess that a system exhibits critical behaviour, one must identify an order parameter going continuously to zero at a critical value of a control parameter. At the critical point, the fluctuations of the order parameter must diverge, as well as the range of temporal and spatial correlations. In finite systems criticality implies that the cutoff in power law behaviour should scale with the system size.

A classic example of a second order phase transition is the ferro-paramagnetic transition exhibited by the Ising model Hamiltonian in zero external magnetic field. The magnetization per spin, which plays the role of the order parameter, is different than zero (the ferromagnetic phase) at low temperature, namely below the critical temperature $T_c$, vanishes at $T_c$ and remains zero above $T_c$ (the paramagnetic phase). For this transition the response function is the magnetic susceptibility, namely the derivative of the magnetization with respect to a vanishing external magnetic field. This susceptibility is indeed the spatial integral of the fluctuations of the magnetization and diverges at $T_c$ due to the divergence of the correlation length, namely the size of clusters of correlated spins. All these singularities approaching $T_c$ behave as power laws, defining the well-known Ising model universality class [28]. It is interesting to mention that the Ising model, besides this second order phase transition in zero field, also shows a first order phase transition for varying non-zero magnetic fields, below $T_c$. A first order phase transition is conversely characterized by a discontinuous order parameter and, eventually, the presence of hysteresis [29]. Increasing fluctuations approaching the transition can be observed also in this context, but the scaling with the system size is usually lacking, hampering the definition of critical exponents. The choice of the control parameters is, therefore, crucial in determining the order of the phase transition and the two phenomena can coexist in the same model (see Discussion).

This well-established scenario for equilibrium systems described by a Hamiltonian has been extended also to non-equilibrium open systems, even in the case where a Hamiltonian is not defined [30]. In particular, neuronal networks (biological and models) represent an instance of an entire field of systems where non-equilibrium phase transitions can occur in a self-organized matter, namely in the absence of a tuning parameter but due to the interaction of many degrees of freedom [20, 31, 32]. Also in non-equilibrium systems a continuous transition can be observed, with critical exponents typical of a well-defined universality class. More specifically, according to the conjecture by Janssen and Grassberger [30], systems with short-range interactions, exhibiting a continuous phase transition into a single absorbing state, belong generically to the Directed Percolation universality class, provided that they are characterized by a one component order parameter without additional symmetries and without unconventional features such as quenched disorder. Different scaling behavior is expected to occur in systems where at least one of these requirements is not fulfilled. We stress that both

directed percolation and branching process on a tree, therefore in the mean field approximation, do provide the same universality class.

Here we reconsider the stochastic WCM in this framework, addressing the central issue related to its critical behaviour. First, we observe that this model is defined by dynamical equations which are not derived from a Hamiltonian function describing the energy of the system. Therefore a real thermodynamic phase transition, where singularities in the second derivative of the free energy occur, is not expected. However, we present a systematic analysis of the features typical of a critical behaviour, showing that a bona fide critical point in the parameter space of the WCM can be actually identified. In particular, we show that: i) the mean firing rate plays the role of the order parameter, passing from zero value to a finite value across the critical point; ii) the correlation time of the order parameter diverges at the critical point; iii) the avalanche size and duration distributions follow a power-law behaviour; iv) for finite systems, this power-law regime scales with the system size, as expected at the critical point. Moreover, we show that the critical exponents fall within the universality class of the mean-field branching process [13].

## Results

### The stochastic Wilson-Cowan model

The stochastic version of the Wilson-Cowan model [26] describes the coupled dynamics of $N_E$ excitatory and $N_I$ inhibitory neurons. The state $a_i$ of neuron $i$ can be active active ($a_i = 1$) or quiescent ($a_i = 0$) and evolves according to a continuous-time Markov process. The transition rate from an active state to a quiescent state ($1 \rightarrow 0$) is $\alpha$ for all neurons, while the rate for the inverse transition ($0 \rightarrow 1$) is described by the activation function $f(s_i)$ that depends on the $i$-th neuron. The total synaptic input $s_i$ is defined as

$$s_i = \sum_j w_{ij} a_j + h_i, \tag{1}$$

where $w_{ij}$ are the synaptic strengths and the parameter $h_i$ plays the role of a small external input that adds up to the synaptic inputs from the connected neurons and the sum runs over all neurons. The activation function is given by

$$f(s) = \begin{cases} \beta \tanh(s) & \text{if } s > 0, \\ 0 & \text{if } s \leq 0, \end{cases} \tag{2}$$

where $\beta$ has the dimension of an inverse time. The quantity $s$ represents the distance of the membrane potential from the firing threshold and is measured in mV. In Eq (2) $s$ is made a-dimensional by dividing by 1 mV. In the following we consider that each neuron is coupled with all other neurons. The synaptic weights $w_{ij}$ are equal to $w_{EE}/N_E$ for excitatory-excitatory connections, $w_{IE}/N_E$ for excitatory-inhibitory connections, $-w_{EI}/N_I$ for inhibitory-excitatory connections and $-w_{II}/N_I$ for inhibitory-inhibitory connections. Therefore the input of a neuron, in the large $N$ limit, only depends on the excitatory or inhibitory type of the neuron, namely $s_i = s_E$ if the $i$-th neuron is excitatory, and $s_i = s_I$ if the $i$-th neuron is inhibitory. In the following we set $\alpha = 0.1 \text{ ms}^{-1}$, $\beta = 1 \text{ ms}^{-1}$, $N_E = N_I = N$, $h_E = h_I = h$, and consider symmetric synaptic weights $w_{EE} = w_{IE} = w_E$, $w_{II} = w_{EI} = w_I$. Thus $s_E = s_I = s$, with $s = \frac{w_E}{N} k - \frac{w_I}{N} l + h$, and $0 \leq k \leq N$ and $0 \leq l \leq N$ are respectively the numbers of active excitatory and inhibitory

neurons [26]. In the following we will focus on the instantaneous firing rate, defined as

$$R = \left(1 - \frac{l+k}{2N}\right) f(s),\tag{3}$$

so that the mean number of neurons that fire in a small time interval $\Delta t$ is given by $NR\Delta t$.

The temporal evolution of the system can be effectively described in terms of the coupled non-linear Langevin equations [33]

$$\frac{dk}{dt} = -\alpha k + f(s)(N-k) + \sqrt{\alpha k + f(s)(N-k)}\,\eta_E(t), \quad (4a)$$

$$\frac{dl}{dt} = -\alpha l + f(s)(N-l) + \sqrt{\alpha l + f(s)(N-l)}\,\eta_I(t), \quad (4b)$$

where the noises satisfy $\langle \eta_i(t) \rangle = 0$, $\langle \eta_i(t)\eta_j(t') \rangle = \delta_{ij}\,\delta(t-t')$. Following Ref. [26], we make a Gaussian approximation and set that the number of active neurons is the sum of a deterministic component and a stochastic perturbation, i.e. $k = NE + \sqrt{N}\xi_E$ and $l = NI + \sqrt{N}\xi_I$. Introducing the variables $\Sigma = (E+I)/2$ and $\Delta = (E-I)/2$, which represent the total average activity and the imbalance between excitatory and inhibitory activity, respectively, and expanding Eq (4) in powers of $N^{-1/2}$, the leading terms proportional to $N$ provide a set of dynamical equations for the deterministic components

$$\frac{d\Sigma}{dt} = -\alpha\Sigma + (1-\Sigma)f(s), \quad (5a)$$

$$\frac{d\Delta}{dt} = -[\alpha + f(s)]\Delta. \quad (5b)$$

At long times, $\Delta$ relaxes to the fixed point value equal to zero, expression of the balance of excitation and inhibition [34] and direct consequence of the hypothesis of symmetric synaptic connections. Conversely, $\Sigma$ relaxes to the fixed point $\Sigma_0$, given by the solution of the equation

$$\alpha\Sigma_0 = f(s_0)(1-\Sigma_0),\tag{6}$$

with $s_0 = w_0\Sigma_0 + h$ and $w_0 = w_E - w_I$. We stress that, by this definition, $w_0$ expresses the relative balance between the excitatory and inhibitory connection strength, and it will turn out to be the parameter controlling the critical transition. In addition, terms proportional to $N^{1/2}$ in Eq (4), can be written as the linearized Langevin equations for the fluctuating components [22, 25, 26]

$$\frac{d}{dt}\begin{pmatrix} \xi_\Sigma \\ \xi_\Delta \end{pmatrix} = \begin{pmatrix} -1/\tau_1 & w_{\rm ff} \\ 0 & -1/\tau_2 \end{pmatrix}\begin{pmatrix} \xi_\Sigma \\ \xi_\Delta \end{pmatrix} + \sqrt{\alpha\Sigma_0}\begin{pmatrix} \eta_\Sigma(t) \\ \eta_\Delta(t) \end{pmatrix},\tag{7}$$

where $\xi_\Sigma = (\xi_E + \xi_I)/2$, $\xi_\Delta = (\xi_E - \xi_I)/2$, the feed-forward term $w_{\rm ff} = (1-\Sigma_0)(w_E + w_I)f'(s_0)$ and

$$1/\tau_1 = \alpha + f(s_0) - (1-\Sigma_0)w_0 f'(s_0), \quad (8a)$$

$$1/\tau_2 = \alpha + f(s_0). \quad (8b)$$

The times $\tau_1$ and $\tau_2$ represent the correlation times in the linear approximation of the dynamical equations. Indeed, in such approximation the temporal correlation functions $C_{xy}(t) = \langle x(t)y(0) \rangle - \langle x \rangle\langle y \rangle$, where $x$ and $y$ are two observables and where the symbol $\langle \cdots \rangle$ represents an average over noise in the stationary state (see Methods), can be written as the linear

combinations of two exponential decays [35] (see Methods for the explicit expressions)

$$C_{xy}(t) = A_{xy,1} e^{-t/\tau_1} + A_{xy,2} e^{-t/\tau_2}. \tag{9}$$

Note that Eq (6) can have more than one solution. In this case, the relevant one is the one characterized by positive values of the relaxation times $\tau_1$ and $\tau_2$, so that the fixed point is attractive. On the other hand, fixed points characterized by negative values of either $\tau_1$ or $\tau_2$ are repulsive and not relevant to the dynamics of the system. An accurate analysis of the stability properties of the WCM for finite external fields can be found for instance in [36].

### Critical point of the dynamical equations

We here discuss the behavior of the system predicted by the linear noise approximation, namely in the limit of very large system size. When the external inputs $h$ is zero, $\Sigma_0 = 0$ is always a solution of the fixed point Eq (6). However, one can show, by taking the linear approximation of the hyperbolic tangent, that there is a critical value $w_{0c} = \alpha\beta^{-1}$. This value expresses the balance between the activation and disactivation characteristic neuronal times and therefore can be interpreted as an optimal value for excitation/inhibition balance. In particular, for $w_0 < w_{0c}$ (when inhibition dominates) the fixed point $\Sigma_0 = 0$ is stable, whereas for $w_0 > w_{0c}$ (when excitation dominates) it is unstable and another stable point $\Sigma_0 \simeq (w_0 - w_{0c})/w_0 > 0$ appears continuously from zero at the onset of the transition. When $h > 0$ there is always only one attractive fixed point with $\Sigma_0 > 0$ and the transition is smoothed out (see Methods).

In Fig 1A we show the firing rate $R_0 = (1 - \Sigma_0)f(w_0\Sigma_0 + h)$ computed at the attractive fixed point, as a function of $w_0$ for different external input $h$. This quantity shows the typical behaviour of an order parameter. In particular, for $h = 0$, $R_0 = 0$ for $w_0 < w_{0c}$, whereas it continuously increases for $w_0 \geq w_{0c}$ as $R_0 \sim (w_0 - w_{0c})$, according to what expected in a second-order phase transition. For finite $h$ values, $R_0$ shows a qualitatively similar behaviour characterized by a continuous increase, and the transition is smoothed out.

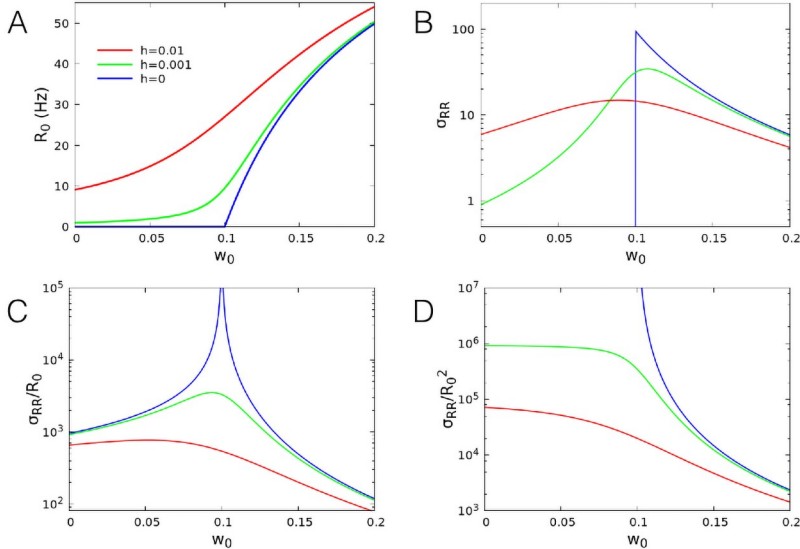

**Fig 1. Order parameter and its variance.** (A) Analytical dependence of the firing rate per neuron at the fixed point on the value of $w_0$, for different values of $h$. (B) Normalized variance $\sigma_{RR} = N\langle(R - R_0)^2\rangle$ as a function of $w_0$. (C) Fano factor $\sigma_{RR}/R_0$. (D) Square coefficient of variation $\sigma_{RR}/R_0^2$, that is equal to $N$ times the variance of the ratio $R/R_0$. Other parameters: $\alpha = 0.1$ ms$^{-1}$, $\beta = 1$ ms$^{-1}$, $w_E + w_I = 13.8$.

Next we analysed the behaviour of the variance of the firing rate at the fixed point, as a function of $w_0$ for different $h$ values. For a large number of neurons, the variance of the firing rate is proportional to $N^{-1}$ (see Methods), therefore $\sigma_{RR} = N\langle(R - R_0)^2\rangle$ is independent of $N$, and can be computed in the linear approximation (see Methods). In Fig 1B we show $\sigma_{RR}$ as a function of $w_0$ for several values of $h$. For $h = 0$ it reaches a maximum at the critical point $w_{0c} = 0.1$ and sharply vanishes for $w_0 < w_{0c}$. For finite values of $h$, the variance shows a smooth maximum close to the critical point. The fact that the variance of the order parameter does not diverge at the critical point, unusual in the framework of second order phase transitions, can be attributed to the vanishing of the noise amplitude in Eq (7). This is due to the particular choice of the activation function. Indeed, different functional forms for $f(s)$ lead to the a non-zero $\Sigma_0$ at the critical point and to diverging fluctuations. However, such a divergence is observed in the ratio of the variance to the mean value of the order parameter. In Fig 1C we show the Fano factor of the firing rate, that is the ratio $\sigma_{RR}/R_0$. This quantity is defined as the ratio of the variance and the mean value, and measures how much the statistics of a variable deviates from the behaviour expected for a Poissonian variable. In the present case, it diverges at the critical point for $h = 0$, with a behaviour $\sigma_{RR}/R_0 \sim |w_0 - w_{0c}|^{-1}$, while it shows a maximum near $w_0 = w_{0c}$ for $h > 0$.

Moreover, in Fig 1D we show the squared coefficient of variation $\sigma_{RR}/R_0^2$. Considering that the linear approximation is derived under the condition that fluctuations are much smaller than the average firing rate (in this case close to the fixed point value), this quantity can be interpreted as the limiting value of $N$ for its validity. Indeed, if $N \gg \sigma_{RR}/R_0^2$, then the standard deviation $\sqrt{\langle(R - R_0)^2\rangle}$ is much smaller than the mean $R_0$ and the linear approximation holds (see below), conversely for $N \ll \sigma_{RR}/R_0^2$ the opposite is true. Therefore, the divergence of this quantity near the critical point means that, no matter how large $N$ is, the linear approximation does not apply.

The critical behaviour in standard critical phenomena is accompanied by the slowing down of the dynamics. This is evidenced by the divergence of the characteristic time-scales of the system. To study the decay of the correlation function close to the critical point, we observe that when $h = 0$ and $w_0 \simeq w_{0c}$, $\Sigma_0$ and $s_0 = w_0\Sigma_0$ are much smaller than one, so that $\tanh(s_0) \approx s_0$. Using this approximation in Eq (8), we find that $\tau_1 \sim \frac{1}{\beta|w_0 - w_{0c}|}$ both for $w_0 < w_{0c}$ and $w_0 > w_{0c}$, while $\tau_2 \le \alpha^{-1}$. If $h > 0$, the divergence of $\tau_1$ is rounded up, and one finds a maximum at $w_0 = w_{0c}$, diverging for $h \to 0$. In Fig 2A and 2B we show the autocorrelation time $\tau_1$ in the linear approximation, where a clear divergence is observed for zero external field.

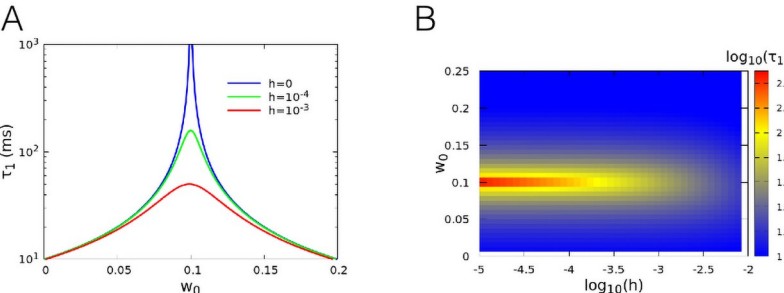

**Fig 2. Divergence of the correlation time at criticality.** (A) Analytical result for the decay time $\tau_1$ in the linear approximation, for the same parameters of Fig 1. (B) Contour plot of $\tau_1$ as a function of both $h$ and $w_0$. The graphs show the divergence of the decay time at the critical value $w_{0c} = 0.1$.

In conclusion, near the critical point $w_0 = w_{0c}$ and for $h = 0$, both the Fano factor and the correlation time of the firing rate diverge. Divergence is also found for the Fano factor and correlation time of other dynamical variables, e.g. the total number of active neurons. These results provide further evidence that the occurring phenomenology can be rightfully interpreted in the framework of critical systems.

## Firing rate dynamics

Near the critical point the linear approximation does not hold even for very large system sizes. We support this conclusion by analysing the instantaneous firing rate (Fig 3) as a function of time for $h = 10^{-5}$ and three values of $w_0$, $w_0 = 1$ (upper row), $w_0 = 0.2$ (middle row) and $w_0 = 0.1$ (lower row). For each value of $w_0$ we show two values of $N$, $N = 10^3$ on the left and $N = 10^5$ on the right. For $w_0 = 1$ (upper row) the normalized variance is $\sigma_{RR} \simeq 6$, so that the dynamics for $N \gg 6$ is always "continuous", smoothly fluctuating around the attractive fixed point, and can be accurately described within the linear approximation. For $w_0 = 0.2$ (middle row), the normalized variance is $\sigma_{RR} \simeq 2400$, therefore if $N < \sigma_{RR}$ (left) the dynamics of the system is irregular and characterized by avalanches. The firing rate frequently hits the value $R = 0$, and a "downstate" of the network follows, where the activation of the neurons is controlled only by the external input $h$ and the activity recovery can take a long time if $h$ is small. On the right, conversely, $N \gg \sigma_{RR}$, and the dynamics becomes "continuous", as in the previous case. Finally, near the critical point, for $w_0 = 0.1$ (lower row), the normalized variance is $\sigma_{RR} \simeq 4.6 \times 10^7$, therefore the dynamics is characterized by avalanches up to $N = 10^5$ and above.

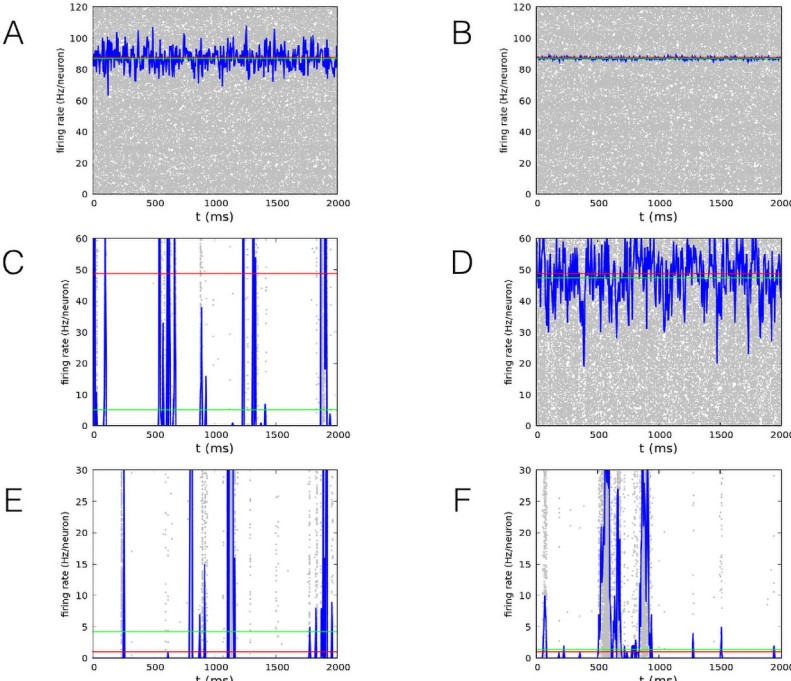

**Fig 3. Firing rate for neuron at the critical point and far from it.** Firing rate measured in numerical simulations as a function of time for $w_E + w_I = 13.8$, $h = 10^{-5}$. Upper row: $w_0 = 1$ (E dominates) (A), middle row: $w_0 = 0.2$ (E dominates) (C), lower row: $w_0 = 0.1$ (E/I balance) (E), left column: $N = 10^3$, right column: $N = 10^5$ (B,D,F). Blue lines represent the firing rate of the network, while gray dots represent single neuron spikes. Red lines show the value of the firing rate $R_0$ at the fixed point of the dynamics. Note that this value can be quite different from the mean firing rate (green lines), when large non linear effects are present.

This qualitative analysis suggests that the occurrence of the avalanche activity in dynamics of the stochastic WCM is indeed related to the presence of a critical point. If the system is moved away from the critical point, this kind of behaviour persists as long as the size of the system is small enough and disappears for larger sizes. More precisely, the system size must be smaller than the squared coefficient of variation of the firing rate. In this case, fluctuations of the firing rate are much larger than the mean value and the dynamics becomes avalanche-like.

## Avalanche dynamics

The above conclusions are strengthened by the quantitative analysis of the avalanche dynamics. We study the distribution of avalanches in the WCM simulated by the Gillespie algorithm [37] (see Methods). We implement two different procedures to define an avalanche and we start discussing the statistics of avalanches defined by the discretization in time bins of the temporal signal. More precisely, we divide the time in discrete bins of width $\delta$ [9] and identify an avalanche as a continuous series of time bins in which there is at least one spike (i.e., a transition of one neuron from a quiescent to an active state). The size of the avalanche is defined as the total number of spikes, while the duration is the number of time bins of the avalanche multiplied by the width $\delta$ of the bins.

In Fig 4 we show the dependence of the size and duration distribution functions on the time bin $\delta$, for $w_0 = 0.1$, $h = 10^{-6}$, $N = 10^6$. The behaviour of the distribution for small and large

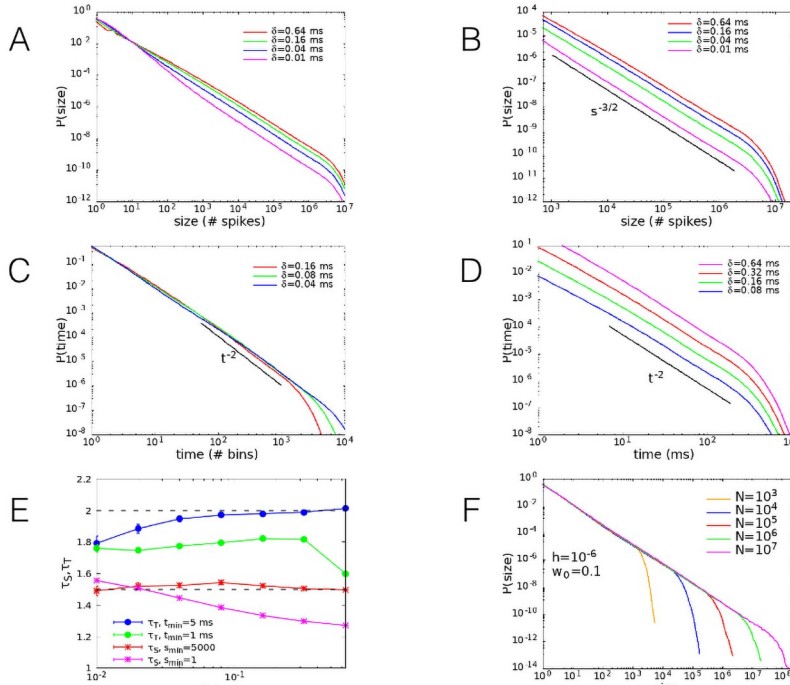

**Fig 4. Size and duration avalanche distributions.** (A) Distribution function of the avalanche sizes on the whole observed range. (B) Distribution function of the avalanche sizes on the region where robust power law behaviour is observed. (C) Distribution function of the avalanche duration as a function of the number of bins. (D) Distribution function of the avalanche duration as a function of time. Parameters $w_0 = 0.1$, $h = 10^{-6}$, $N = 10^6$. Different curves correspond to different values of the bin width $\delta$, introduced to define the avalanche (see Methods). (E) Exponents of size and duration distributions, with error bars, computed using the estimator introduced in [38, 39], for different values of the lower bound of the fitting window. Error bars are not shown if they are smaller than the symbol size. (F) Size distribution function of the avalanches for $w_0 = 0.1$, $h = 10^{-6}$, and different values of the system size $N = 10^3 - 10^7$. As expected for a critical behaviour in finite systems, the exponential cut-off scales with $N$.

avalanche sizes is separately evidenced in Fig 4A and 4B. We notice that at small sizes the slope of the curves strongly depends on the bin width, as well evidenced also in experimental data [9–12]. Conversely, at large sizes all the curves exhibit a slope quite independent of the bin width, according to the power-law dependence $P(S) \sim S^{-\tau_S}$ with an exponent $\tau_S$ very close to 3/2. Analogously, in Fig 4C and 4D the distribution of avalanche durations exhibits a scaling $P(T) \sim T^{-\tau_T}$ with an exponent $\tau_T \sim 2$, which is very robust with the bin width. In Fig 4E we show the values of both exponents with error bars that best fit the data using the estimator introduced by Clauset et al. [38] (see Methods). It is evident that, if exponents are evaluated restricting the procedure to the large avalanche regime, their value converges to the expected exponents of the mean field branching process universality class, independently of the bin size [13]. This observation suggests that an underlying mechanism of marginal propagation of neural activity could be responsible for the avalanche behaviour.

As expected for critical phenomena in finite systems, at the critical point the power-law behaviour of the avalanche size distribution function presents an exponential cut-off that scales with the system size. This is clearly shown in Fig 4F for $w_0 = 0.1$ and vanishing external field, confirming that the power-law behaviour of the distribution is a genuine expression of the absence of a characteristic size at the critical point. We have also considered a different avalanche definition, through the introduction of a finite threshold in activity [40], confirming that this can lead to wrong values of critical exponents [41] (see Methods).

In recent years, the scaling properties of the avalanche shape have received wide attention in the community [42], searching for the collapse onto a universal curve according to a specific rescaling. Indeed, this analysis has been first proposed in the context of the crackling noise [43], where the scaling exponent for the avalanche size as a function of its duration has been derived to be $\gamma = (\tau_T - 1)/(\tau_S - 1)$. Under this assumption, it is possible to obtain the collapse of the shapes of avalanches with different sizes onto a universal curve. Here we examine first the scaling behavior of the avalanche size versus its duration, see Fig 5A and 5C. Results evidence the expected scaling behavior of the avalanche size vs. its duration: The exponent $\gamma$ is slightly larger than 2, the value predicted by the previous relation for $\tau_T = 2$ and $\tau_S = 1.5$ ($\gamma \sim 2.1 \pm 0.05$ for 50% inhibitory neurons and $\gamma \sim 2.02 \pm 0.05$ for 20% inhibitory neurons, where the fit is on the interval $T \in [80, 200]$). Moreover, the avalanche shapes for different sizes collapse onto a universal function for durations up to 400 time steps, corresponding to the scaling regime for the size distributions. However, at variance with the results from crackling noise [43], the shape is non-parabolic and strongly asymmetrical for all avalanche durations, for both percentages of inhibitory neurons.

## Temporal correlation functions

In order to complete the description of the critical behaviour shown by the WCM, we focus here on the temporal correlation function of the firing rate simulated by the Gillespie algorithm. At each time we compute the mean total activity $\Sigma = (k + l)/2N$ and the difference $\Delta = (k - l)/2N$. We focus on the correlation function of the mean firing rate $R = (1 - \Sigma)f[w_0\Sigma + (w_E + w_I)\Delta + h]$ in the stationary state

$$C_{RR}(t) = \frac{\langle R(t)R(0)\rangle - \langle R(0)\rangle^2}{\langle R(0)^2\rangle - \langle R(0)\rangle^2}. \tag{10}$$

In Fig 6 we show the dependence of the correlation function on the number of neurons $N$ for $h = 10^{-6}$ and two values of $w_0$, $w_0 = 0.2$ far from the critical point and $w_0 = 0.1$, corresponding to the critical point for $h \to 0$. In both cases, the correlation function simulated by the Gillespie algorithm (dots) tends to the value predicted by the linear approximation (continuous

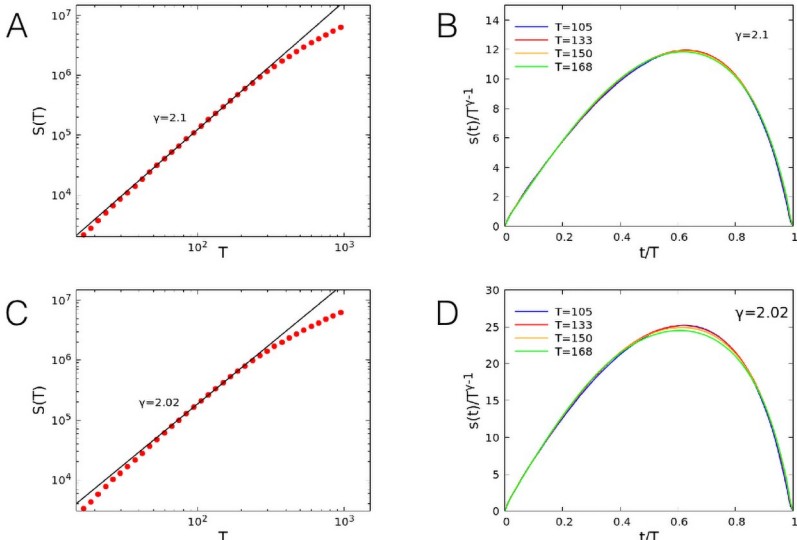

**Fig 5. Shape of the avalanche distributions.** Scaling of the avalanche size $S$ as a function of its duration $T$ for networks with 50% (A) and 20% (C) inhibitory neurons. Collapse of the avalanche shape for avalanche size in the scaling regime for notworks with 50% (B) and 20% (D) inhibitory neurons. Parameters $w_0 = 0.1$, $h = 10^{-6}$, $N = 10^7$.

line) given by Eq (9) for $N \to \infty$. However for $w_0 = 0.2$, far from the critical point, numerical data reproduce the linear approximation as soon as $N \gtrsim 10^5$, while for $w_0 = 0.1$ the convergence is much slower. In Fig 7 we show the dependence of the correlation function on $h$ for a fixed value of the number of neurons, $N = 10^7$. Data confirm that at the critical point ($w_0 = 0.1$) the decay of the correlation function slows down in the limit $h \to 0$. As expected, critical slowing down is not observed for $w_0 = 0.2$.

The maximum correlation time, obtained from an exponential fit of the long time tail of the functions, is plotted as a function of $h$ in Fig 8A for $w_0 = 0.1$. For a fixed value of the number of neurons $N$, the correlation times saturate at a finite value at the critical point $w_0 = 0.1$. The value at which the time saturates however increases with the system size, so that the range of agreement of the measured correlation time with the linear approximation prediction extends toward smaller values of $h$ for increasing $N$. In the limit $N \to \infty$ the correlation time is always given by the linear approximation for any value of $h$, and therefore diverges for $h \to 0$. In Fig 8B we plot the maximum correlation time as a function of $N$ for $h = 10^{-6}$ and $w_0 = 0.1, 0.2$. It

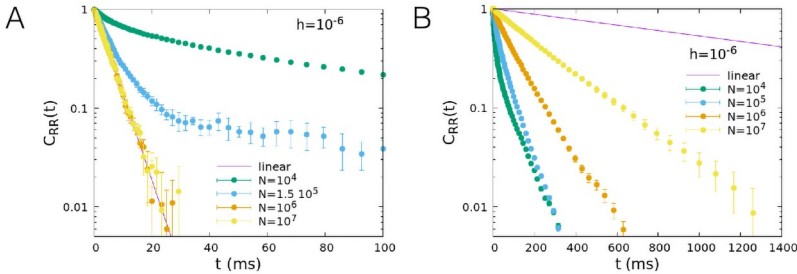

**Fig 6. Temporal decay of the firing rate autocorrelation, role of the system size $N$.** Time correlation function of the firing rate for several values of $N$, for $\alpha = 0.1$ ms$^{-1}$, $\beta = 1$ ms$^{-1}$, $w_E + w_I = 13.8$, $h = 10^{-6}$, and $w_0 = 0.2$ (A), $w_0 = 0.1$ (B). Dots correspond to the correlation function of the model simulated with the Gillespie algorithm, while the continuous line corresponds to the linear approximation, that is valid for large values of $N$.

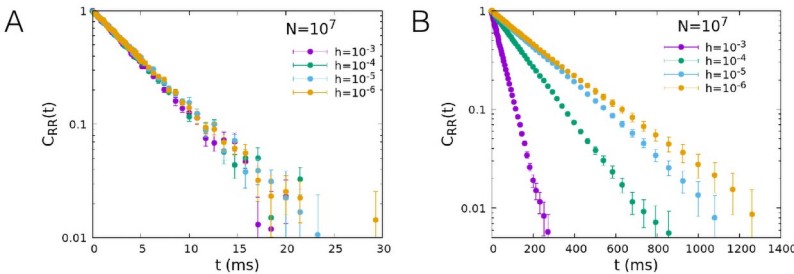

**Fig 7. Temporal decay of the firing rate autocorrelation, role of the external input $h$.** Time correlation function of the firing rate for several values of $h$, for $N = 10^7$ and $w_0 = 0.2$ (A), $w_0 = 0.1$ (B), other parameters as in Fig 6.

can be observed that the relaxation time saturates to the large value predicted by the linear approximation for $N \to \infty$ at the critical point $w_0 = 0.1$, whereas it decreases to smaller values far from the critical point.

Our analysis of the firing rate correlation function provides further evidence of the critical behaviour occurring in the WCM. In particular, the divergence of the characteristic time is in agreement with the slowing down of the dynamics in systems close to the critical point.

## Discussion

The origin and nature of the power-law behaviour of the spontaneous activity in neural systems is a long-standing open issue. The observation of this dynamics in real systems is widespread, as well as in different models proposed to explain it. Similar scaling behaviour is indeed observed in a variety of integrate and fire neuronal network models, either self-organized, i.e. in absence of a tuning parameter [17, 20], or by adjusting at an appropriate value a relevant parameter [14, 19]. The central question in this context is whether scale-free phenomena are the mirror image of a genuine critical behaviour or emerge from non linear stochastic dynamics. Among different approaches, the stochastic Wilson-Cowan model, formulated in terms of the activity of populations of neurons, describes many interesting phenomena observed in neural dynamics. Moreover, it presents the advantage of the possibility of being analysed not only numerically but, most importantly, by an analytical approach under certain approximations. In previous studies this model was indicated as an example where the emergence of neuronal avalanches in the activity is not associated with a genuine critical point, but rather the byproduct of the network structure with noisy neuronal dynamics [26]. In order to clarify this point, we propose to explore a wider range of the parameter space, focusing on the

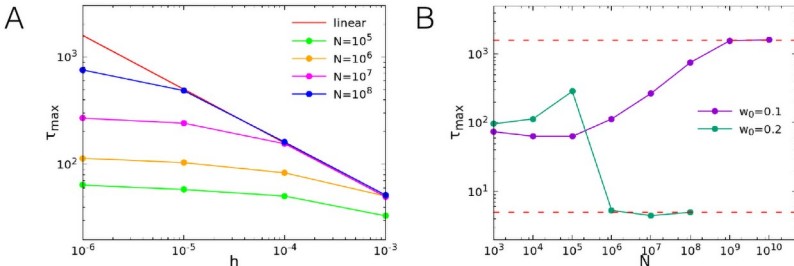

**Fig 8. Long time scale behaviour as a function of the system size $N$ and the external input $h$.** (A) Maximum correlation time extracted from an exponential fit of the long time tail of the correlation function, as a function of $h$ and for different values of $N$, for $w_0 = 0.1$. The continuous red line corresponds to the linear approximation. (B) Maximum correlation time as a function of $N$ for $h = 10^{-6}$ and $w_0 = 0.1, 0.2$.

behaviour of different quantities, as the temporal correlation function of the firing rate, in order to verify if a critical point can be identified, shedding new light on the nature of the phenomenon observed in the model.

Several papers in previous literature have studied the WCM and similar models, evidencing different kinds of behaviors, as first order phase transitions at the bifurcation point [36, 44, 45]. For instance, in [36], the authors considered the bifurcation transitions appearing upon varying the (always finite) external voltage inputs. However, the observed first-order transitions occur in a region of the parameter space of the model which does not overlap with that studied in our work. In particular, the theoretical results show slowing down and increasing fluctuations close to the bifurcation point and describe real systems in specific conditions, such as anesthesia, sleep cycles or seizures (see for instance [46]), very differently from the spontaneous activity state analyzed in the present study, namely in the absence of strong external stimulations, such as administration of anesthetic drugs, neither to the case of transitions between different sleep states, nor to the situation where epileptic crises can occur. In the review by Breakspear and co-workers [47], the possible different behaviors in brain activity are discussed. It is shown how the thermodynamic phase transitions in spatially extended systems are somehow the counterpart of the bifurcation transitions for systems with few components. This is an interesting remark, since second order phase transitions are observed only in systems with a sufficiently large number of degrees of freedom, namely in the thermodynamic limit.

In the light of the above discussion, the first important remark is that critical behaviour is expected for vanishing external fields. This requirement, together with the limit of very large system sizes, represents the foundation of the symmetry breaking phenomenon originating second order phase transitions. Recently critical behavior has been observed in non-zero field in the presence of self-adaptation mechanisms [20, 48]. Here we analyse the analytical solution of the WCM for vanishing $h$ and in a wide range of the parameter values $w_0$, searching for a quantity playing the role of an order parameter. In fact, the linear noise approximation is derived [22, 26] in the limit of very large $N$, namely in the thermodynamic limit. The analytical solution evidences that it exists a particular value of the control parameter $w_{0c}$ above which, for vanishing $h$, a second fixed point appears, beside the absorbing state $\Sigma_0 = 0$. In the neighbourhood of this critical value $w_{0c}$, the system activity is small allowing for the linearisation of the activation function. Interestingly, the critical value $w_{0c} = \alpha\beta^{-1}$ is the ratio of two characteristic rates, the disactivation rate $\alpha$ and the activation one, $\beta$. Criticality is therefore tuned by the optimal balance between these two (active-inactive) transitions: Below $w_{0c}$ the disactivation rate is much shorter than the activation rate and the absorbing state attracts the dynamics. Conversely, for very short activation times global activity becomes self-sustained even in the absence of external fields. An intriguing alternative interpretation of the expression of the critical point stems from the consideration that $\beta w_{0c} = \alpha$ is the characteristic neuronal disactivation time. The l.h.s. is reminiscent of the product $\beta J$ in the Ising model, where $\beta$ is the inverse temperature (entropic parameter) and $J$ the interaction strength (energetic parameter). Therefore, the value of the critical point expresses also a non-local balance between the strength of the connection between pre- and post-synaptic neuron and the activation time in the post-synaptic neuron.

Within the linear approximation hypothesis, the analytical solution is able to provide a coherent description of the system dynamics in terms of the second order phase transition framework. More precisely, the system firing rate plays the role of the order parameter, going to zero at $w = w_{0c}$, the correlation time diverges, evidencing the critical slowing down, as well as the fluctuations of the firing rate with respect to the fixed point value. Interestingly, the fixed point value for the variable $\Delta$ is $\Delta_0 = 0$, independently of $w_0$. This suggests that the

balance of the activity of excitatory and inhibitory neurons is a necessary but not sufficient condition for criticality. However, the dynamic Eq (5) are derived under the hypothesis of equal size populations of excitatory and inhibitory neurons, as well as symmetric connection strengths between different populations. Therefore, in order to further investigate in the WCM the role of balance of excitation and inhibition on the activity critical properties, it is necessary to extend the analytical study and generalize the analytical solutions relaxing the above hypothesis.

This analysis offers then a coherent scenario to understand the WCM behaviour and provides as well a tool to infer in which limit the linear approximation fails giving raise to bursty behaviour. More precisely, in order to observe neuronal avalanches, the coefficient of variation, measuring the fluctuations of the order parameter, should be much larger than the system size. In particular, for $w_0 \gtrsim w_{0c}$ (Fig 3C and 3D) neuronal avalanches are found for the smaller system size, whereas the linear approximation (continuous behaviour) holds for the larger $N$. At the critical point (Fig 3E and 3F), fluctuations are larger than any $N$ and avalanches are always detected. Analogously, far from the critical point (Fig 3A and 3B) the linear approximation always holds and avalanches are never found. Interestingly, similar results have been recently found for a different model [49]. Numerical data for the cortical branching model have evidenced, in fact, that the firing rate goes to zero for a specific value of the control parameter (the branching parameter), where the susceptibility diverges as well. As expected, moving away from the critical point these features are no longer found since the system does not satisfy any more the condition of vanishing external fields and the behaviour ceases to be critical.

Having clarified under which conditions critical behaviour is to be expected, we address next the issue of the scaling behaviour of neuronal avalanches. The determination of critical exponents in experimental systems represent an important challenge in terms of the appropriate tools to identify each avalanche. Common approaches implement the discretization of the temporal signal in bins. This approach leads to exponents varying with the bin size, as a consequence the optimal bin is identified with the one leading to a branching ratio equal to one, signature of a critical branching process [12]. Alternatively, a threshold in the amplitude of the signal can be chosen, defining as avalanche size the area delimited by the signal above threshold. A recent study has shown that special care must be taken implementing this method, in order to get the right critical exponent values [41]. In the present study we implement both approaches to identify the avalanche size, in order to verify the existence and robustness of the universal scaling behaviour. Numerical simulations by the Gillespie algorithm are very efficient numerically and allow the study of very large system sizes. This advantage turned up to be crucial in identifying an interesting scaling behaviour of the distributions for very large avalanche sizes. More precisely, by monitoring the value of the exponents as function of the bin size, without imposing any additional requirement, we evidence that, as expected, $\tau_S$ and $\tau_T$ depend on the bin size for moderate values of $S$, however for large avalanche sizes, $S > 10^3$, the scaling of the distribution becomes independent of $\delta$. This surprising result, evidenced only because the analysis explored seven decades of $N$ values, is in line with what expected from critical phenomena. Indeed, the critical slowing down at the critical point implies the absence of a characteristic time. By rescaling the time variable by a finite $\delta$ the temporal signal should be self-similar and therefore $\delta$-independent. Most importantly, the extension of the scaling regime correctly scales with the system size. This behaviour can be then considered as a further confirmation of the critical nature of the activity in absence of external fields.

The universality class of the scaling behaviour for large avalanche sizes is in agreement with the mean field branching model universality class. These results are consistent with a variety of experimental data on different neuronal systems [9–12] and numerical simulations on

complex networks in finite dimensions [17, 19]. We stress that it could appear surprising the emergence of a mean field universality class in finite dimension. A possible explanation is the small-world feature of functional networks, well supported by a number of experimental results. Moreover, recently it has been shown [50] that, starting from a regular square lattice, a neuronal integrate and fire model exhibits a crossover from the 2d sand pile behavior to the mean field branching process universality class. This crossover is due to the interplay between synaptic plastic adaptation and refractory time which makes the regular lattice evolve into a tree with negligible loops. In order to clarify this point in the context of the WCM we performed preliminary simulations on a 2d square lattice $32 \times 32$. On each site of the lattice we placed one excitatory and one inhibitory neuron, for a total number of neurons $N = 2048$. Each neuron can establish an average number of 80 synaptic connections at random, and the connection probability is proportional to $\exp(-r/5)$, where $r$ is the distance between two neurons measured in lattice constant. Preliminary results show that the avalanche activity exhibits exponents very close to the values detected for the fully connected network. However, due to the limited system size, the scaling regime is limited to about one decade and the estimation of the exponents is not fully accurate. Moreover, in this calculation we implemented the value of the critical point $w_{0c}$ found for the fully connected network, even if a more precise identification is necessary since the critical point is not a universal quantity but depends on the network structure. Therefore, we plan to investigate the WCM behavior in finite dimensions in more detail and with a better statistics in a future study.

Moreover, we observe that an alternative definition of avalanches, implementing a threshold in the signal amplitude, can indeed lead to wrong exponent values (see Methods). In fact, we recover the expected behaviour only for vanishing threshold, whereas for finite thresholds the signal behaves as an Ornstein-Uhlenbeck process. This is an interesting observation since it suggests that neglecting regions in the signal with small amplitudes provides a signal typical of an uncorrelated process, in contrast with the feature of the whole neural activity.

The analytical calculation of the firing rate correlation function confirms the existence of a critical value for the control parameter, $w = w_{0c}$, where the correlation time diverges for vanishing external fields in the linear approximation. Numerical simulations confirm that at the critical point the correlation function tends to the linear approximation behaviour in the limit of very large $N$, with a correlation time which remains finite far from $w_{0c}$ and increases with $N$ at the critical point. This temporal relaxation behaviour, evidence of the critical slowing down, confirms the critical features of the firing rate activity. In conclusion, we confirm that the WCM model, able to reproduce a variety of complex features of neuronal activity, as oscillations and noisy limit cycles [27], exhibits critical behaviour at a specific value of the tunable parameter in the thermodynamic limit and for vanishing external fields.

## Methods

### Stability of the fixed points

From Eq (6) one has that, for $h = 0$, the fixed point $\Sigma_0 = 0$ is unstable if $w_0 > w_{0c}$ and stable if $w_0 < w_{0c}$, where $w_{0c} = \alpha/\beta$. Near the transition $h = 0$, $w_0 = w_{0c} = \alpha/\beta$, assuming both $h$ and $\Sigma_0$ are small, so that a linear approximation of the hyperbolic tangent can be considered, Eq (6) becomes

$$\beta w_0 \Sigma_0^2 + (\alpha - \beta w_0 + \beta h)\Sigma_0 - \beta h = 0. \tag{11}$$

The only acceptable solutions are those with $\Sigma_0 \geq 0$. Since the first and third coefficient have opposite sign, for $h > 0$ there is always exactly one acceptable solution. For $h = 0$, there is always a solution $\Sigma_0 = 0$. When $w_0 \leq w_{0c}$ this is the only acceptable solution, while for

$w_0 > w_{0c}$ we have also the solution $\Sigma_0 \simeq (w_0 - w_{0c})/w_0$ at first order in $w_0 - w_{0c}$. To investigate the stability of the fixed point, we have to consider the sign of the eigenvalues of the Jacobian, that is $\tau_1$ and $\tau_2$ given by Eq (8). These can be written also as

$$\tau_1^{-1} = (1 - \Sigma_0)^{-1}[\alpha - \beta w_0(1 - \Sigma_0)^2 + \alpha^2 \Sigma_0^2 w_0/\beta], \quad (12a)$$

$$\tau_2^{-1} = \alpha(1 - \Sigma_0)^{-1}. \quad (12b)$$

Note that the expressions (12) are exact. Therefore, eigenvalue $\tau_2^{-1}$ is always positive. At the fixed point $\Sigma_0 = 0$, $\tau_1^{-1} = \beta(1 - \Sigma_0)^{-1}(w_{0c} - w_0)$, namely it is positive for $w_0 < w_{0c}$ (fixed point is stable) and negative for $w_0 > w_{0c}$ (fixed point is unstable). For $w_0 > w_{0c}$ and at the fixed point $\Sigma_0 \simeq (w_0 - w_{0c})/w_0$, one has $\tau_1^{-1} \simeq \beta(1 - \Sigma_0)^{-1}(w_0 - w_{0c})$ at first order in $w_0 - w_{0c}$, so that $\tau_1$ is positive and the fixed point is stable.

**Alternative definition of avalanches: Role of the threshold.** In order to investigate the robustness of the observed scaling behaviours, we study the avalanche statistics implementing a different definition of avalanche, which is based on the analysis of the continuous temporal signal of the firing rate. We set a fixed threshold value $\Theta$, and define the avalanche as an interval of time in which the firing rate is continuously above the threshold. The duration of the avalanche is the width of the time interval, while the size can be defined in three different ways: 1) as the total number of spikes observed in the time interval; 2) as the integral of the firing rate in the time interval; 3) as the integral of the difference between the firing rate and the threshold value. Definitions 1) and 2) give quite similar results, because the total number of spikes is proportional to the integral of the firing rate apart from small fluctuations. Fig 9A and 9B show the distributions of avalanche size and durations, defined with $\Theta = 0$. We used definition 2 or 3 (they coincide in the case of $\Theta = 0$) to measure the size of the avalanches. The exponents obtained with this procedure fully agree with the ones obtained by temporal binning, and are therefore those of a mean field branching process.

However, as recently pointed out in [41], for continuous-time signals, the introduction of a finite threshold value in the definition of avalanches can lead to different scaling regimes. In order to verify this point, we consider the case of finite thresholds, $\Theta > 0$. In our analysis, we consider as threshold the mean firing rate, $\Theta = \langle R \rangle$. For $w_0 = 0.1$ and $h = 10^{-6}$ the mean firing rates fall within the interval from 0.63 Hz for $N = 10^3$ to 0.54 Hz for $N = 10^6$ (the firing rate at the fixed point is $R_0 = 0.316$ Hz), while for $w_0 = 0.2$ and $h = 10^{-3}$ the mean firing rates take value in the interval from 11 Hz for $N = 10^3$ to 50 Hz for $N = 10^6$ (the firing rate at the fixed point is $R_0 = 50.3$ Hz).

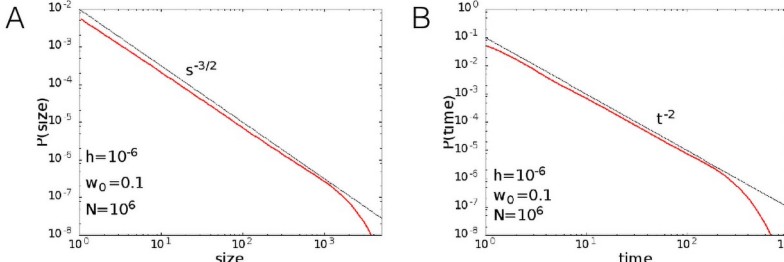

**Fig 9. Avalanche size and duration distributions measured from the analysis of the continuous-time series of the firing ratesignal.** The avalanche is defined as a continuous interval of time in which the firing rate is greater than a zero threshold and its duration is the width of the time interval. (A) Distribution function of the avalanche sizes. (B) Distribution function of the avalanche duration. Parameters: $w_0 = 0.1$, $h = 10^{-6}$, $N = 10^6$. Exponents of size and duration distributions, computed using the estimator introduced in [38, 39] with lower bounds of the fitting windows $S_{min} = 10$, $t_{min} = 10$ ms, are $\tau_S = 1.54 \pm 0.03$, $\tau_T = 2.04 \pm 0.04$.

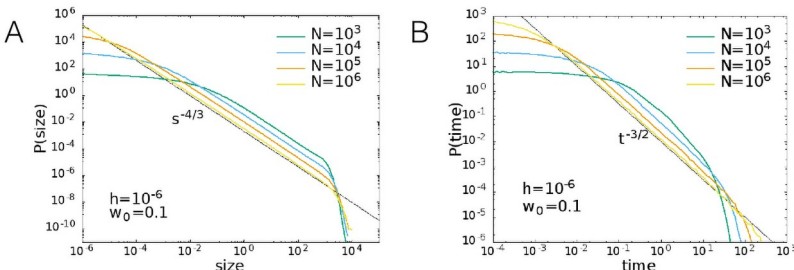

**Fig 10. Avalanche size and duration distributions measured from a finite threshold.** (A) Size distribution and (B) duration distribution according to definition 3 for $w_0 = 0.1$ and $h = 10^{-6}$.

In Fig 10 we show the size and duration distributions for $w_0 = 0.1$ and $h = 10^{-6}$. The sizes were measured with the definition 3, that is as the integral of the difference between the firing rate and the threshold. The observed power-law exponent for the sizes is $-4/3$, while the exponent for the durations is $-3/2$, as expected for a random walk or Ornstein-Uhlenbeck process [41].

Moreover, in Fig 11 we show the size distribution measured according to the definition 2, that is as the integral of the firing rate, therefore integrating also the area below the threshold. In this case, consistently with what observed in [41], one finds also for the size distribution an exponent equal to $-3/2$, as for the duration distribution.

Our analysis confirms the warning resulting from the discussion presented in [41]: The introduction of a threshold can lead to an incorrect evaluation of the scaling behavior. In the present case, choosing the threshold $\Theta = 0$ allows us to recover the same scaling exponents obtained from the alternative definition of avalanche built on the time binning, in the universality class of the mean-field branching process. On the other hand, a different choice of $\Theta$ hides this scaling and reveals a behaviour similar to the simple random walk model.

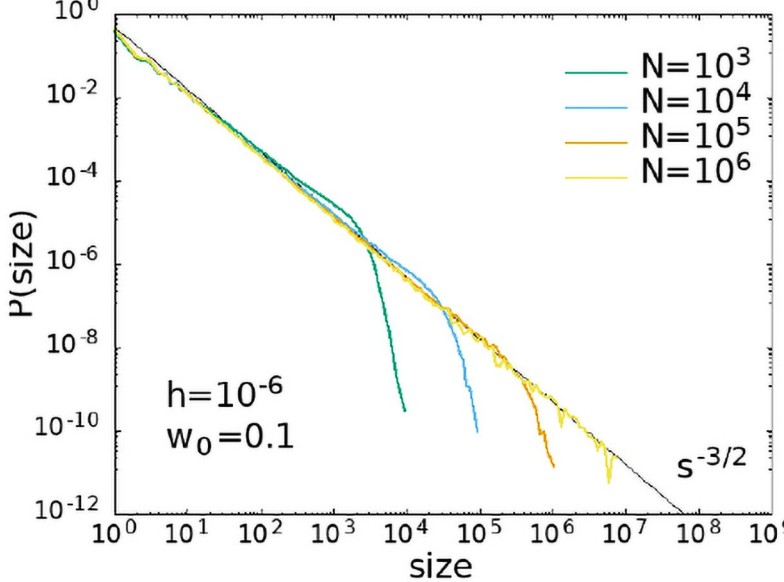

**Fig 11. Avalanche size distributions measured from a finite threshold.** Distribution of the sizes according to definition 2 for $w_0 = 0.1$ and $h = 10^{-6}$.

### Analytic expressions of variances and correlation functions

The covariance matrix $\sigma$ of the system (7) has elements $\sigma_{ij} = \langle \xi_i(0)\xi_j(0)\rangle$ with $i, j = (\Sigma, \Delta)$, where $\langle\cdots\rangle$ denotes an average in the stationary state, which satisfy the relation [51]

$$-\begin{pmatrix} \alpha\Sigma_0 & 0 \\ 0 & \alpha\Sigma_0 \end{pmatrix} = \mathcal{M}\sigma + \sigma\mathcal{M}^T, \tag{13}$$

where $\mathcal{M}^T$ denotes the transpose matrix of

$$\mathcal{M} = \begin{pmatrix} -1/\tau_1 & w_{\mathrm{ff}} \\ 0 & -1/\tau_2 \end{pmatrix}. \tag{14}$$

Solving Eq (13) one obtains

$$\sigma = \frac{\alpha\Sigma_0}{2} \begin{pmatrix} \tau_1\left(1 + \frac{w_{\mathrm{ff}}^2 \tau_1\tau_2^2}{\tau_1+\tau_2}\right) & \frac{w_{\mathrm{ff}}\tau_1\tau_2^2}{\tau_1+\tau_2} \\ \frac{w_{\mathrm{ff}}\tau_1\tau_2^2}{\tau_1+\tau_2} & \tau_2 \end{pmatrix}. \tag{15}$$

The inverse matrix $\sigma^{-1}$ then reads

$$\begin{aligned} \sigma^{-1} &= \frac{2}{\alpha\Sigma_0} \frac{\tau_1 + \tau_2}{2\tau_1\tau_2 + \tau_2^2 + \tau_1^2(1 + w_{\mathrm{ff}}^2\tau_2^2)} \\ &\times \begin{pmatrix} \frac{\tau_1+\tau_2}{\tau_1} & -w_{\mathrm{ff}}\tau_2 \\ -w_{\mathrm{ff}}\tau_2 & \frac{\tau_1+\tau_2+w_{\mathrm{ff}}^2\tau_1\tau_2^2}{\tau_2} \end{pmatrix}. \end{aligned} \tag{16}$$

The elements of the time correlation matrix $\mathcal{C}(t)$ are obtained from the equations [51]

$$\mathcal{C}_{ij}(t) = (e^{\mathcal{M}t}\sigma)_{ij}. \tag{17}$$

The matrix $\mathcal{M}$ has eigenvalues $(-1/\tau_1, -1/\tau_2)$ and eigenvectors $(1, 0)^T$ and $(-w_{\mathrm{ff}}\,\tau_1\,\tau_2/(\tau_1 - \tau_2), 1)^T$. Diagonalizing $\mathcal{M}$, one obtains the matrix exponential

$$\begin{aligned} e^{\mathcal{M}t} &= \begin{pmatrix} 1 & -\frac{w_{\mathrm{ff}}\tau_1\tau_2}{\tau_1-\tau_2} \\ 0 & 1 \end{pmatrix} \begin{pmatrix} e^{-t/\tau_1} & 0 \\ 0 & e^{-t/\tau_2} \end{pmatrix} \\ &\times \begin{pmatrix} 1 & -\frac{w_{\mathrm{ff}}\tau_1\tau_2}{\tau_1-\tau_2} \\ 0 & 1 \end{pmatrix}^{-1} \\ &= \begin{pmatrix} e^{-t/\tau_1} & \frac{\tau_1\tau_2 w_{\mathrm{ff}}(e^{-t/\tau_1}-e^{-t/\tau_2})}{\tau_1-\tau_2} \\ 0 & e^{-t/\tau_2} \end{pmatrix}. \end{aligned} \tag{18}$$

Note that in the case of $\tau_1 = \tau_2$ the upper right element in the above matrix becomes $te^{-t/\tau_1}$.

Next, from Eq (17), one obtains the explicit expressions for the cross-correlation functions

$$
\begin{aligned}
\mathcal{C}_{\Sigma\Sigma}(t) &= \frac{\alpha\Sigma_0\tau_1^2\tau_2^2}{2(\tau_2^2 - \tau_1^2)} \\
&\times \left[(\tau_1^{-1} - \tau_1\tau_2^{-2} - \tau_1 w_{\mathrm{ff}}^2)e^{-t/\tau_1} + \tau_2 w_{\mathrm{ff}}^2 e^{-t/\tau_2}\right]
\end{aligned}
\tag{19}
$$

$$
\mathcal{C}_{\Sigma\Delta}(t) = \frac{\alpha\Sigma_0\tau_1\tau_2^2 w_{\mathrm{ff}}}{2(\tau_1^2 - \tau_2^2)}\left[2\tau_1 e^{-t/\tau_1} - (\tau_1 + \tau_2)e^{-t/\tau_2}\right]
\tag{20}
$$

$$
\mathcal{C}_{\Delta\Sigma}(t) = \frac{\alpha\Sigma_0\tau_1\tau_2^2 w_{\mathrm{ff}}}{2(\tau_1 + \tau_2)}e^{-t/\tau_2}
\tag{21}
$$

$$
\mathcal{C}_{\Delta\Delta}(t) = \frac{\alpha\Sigma_0\tau_2}{2}e^{-t/\tau_2}.
\tag{22}
$$

Within the linear approximation valid in the limit of large number of neurons that we are here considering, and in the stationary state when the deterministic components have relaxed to the attractive fixed point, the firing rate (3) can be written as

$$
R(t) = R_0 + N^{-1/2}\xi_R(t),
\tag{23}
$$

where

$$
\xi_R(t) = R_\Sigma\xi_\Sigma(t) + R_\Delta\xi_\Delta(t),
\tag{24}
$$

and

$$
R_\Sigma = \left.\frac{\partial R}{\partial\Sigma}\right|_{\Sigma_0,\Delta_0} = w_0(1 - \Sigma_0)f'(s_0) - f(s_0) = \alpha - \tau_1^{-1},
\tag{25}
$$

$$
R_\Delta = \left.\frac{\partial R}{\partial\Delta}\right|_{\Sigma_0,\Delta_0} = (w_E + w_I)(1 - \Sigma_0)f'(s_0) = w_{\mathrm{ff}},
\tag{26}
$$

are the derivatives of $R$ computed at the fixed point. The correlation function of $\xi_R(t)$ is therefore given by

$$
\langle\xi_R(t)\xi_R(0)\rangle = R_\Sigma\mathcal{C}_{\Sigma\Sigma}(t) + R_\Sigma R_\Delta[\mathcal{C}_{\Sigma\Delta}(t) + \mathcal{C}_{\Delta\Sigma}(t)] + R_\Delta^2\mathcal{C}_{\Delta\Delta}(t).
\tag{27}
$$

The variance of $R$ can then be computed as $\langle(R - R_0)^2\rangle = N^{-1}\langle\xi_R(0)^2\rangle$ (note the factor $N$ appearing due to the definition 23), and $\sigma_{RR} = N\langle(R - R_0)^2\rangle = \langle\xi_R(0)^2\rangle$, so that

$$
\sigma_{RR} = R_\Sigma^2\sigma_{\Sigma\Sigma} + 2R_\Sigma R_\Delta\sigma_{\Sigma\Delta} + R_\Delta^2\sigma_{\Delta\Delta},
\tag{28}
$$

where $\sigma_{\Sigma\Sigma}$, $\sigma_{\Sigma\Delta}$ and $\sigma_{\Delta\Delta}$ are the elements of the covariance matrix (15).

## Numerical simulation methods

The network dynamics is simulated as a continuous-time Markov process, using the Gillespie algorithm [37]. More precisely, the steps of the algorithm are the following: 1) for each neuron $i$ we compute the transition rate $r_i$: $r_i = \alpha$ if neuron $i$ is active, or $r_i = f(s_i)$ if it is quiescent; 2) we compute the sum over all neurons $r = \sum_i r_i$; 3) we draw a time interval $dt$ from an exponential distribution with rate $r$; 4) we choose the $i$–th neuron with probability $r_i/r$ and change its state; 5) we update the time to $t + dt$.

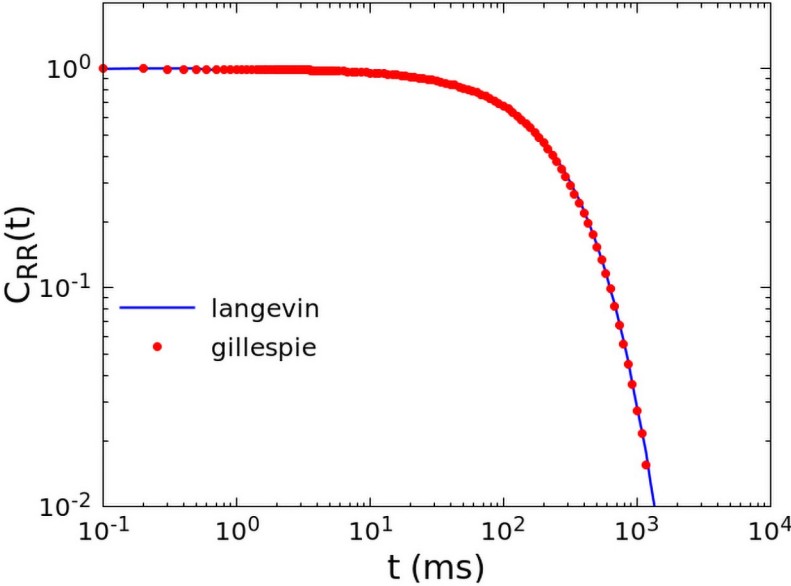

**Fig 12. Equivalence of different numerical simulation methods.** Comparison between the correlation function calculated by the Gillespie algorithm (red dots) and fully non-linear Langevin equation (blue line), for $N = 10^7$, $w_0 = 0.1$, $h = 10^{-6}$.

For a very large number of neurons $N > 10^7$, we optimized the numerical computation by simulating directly the Langevin Eq (4), with a fixed time step $dt = 10^{-3}$ ms. In Fig 12 we show that the two simulation methods coincide perfectly for $N = 10^7$.

## Acknowledgments

LdA and AS acknowledge support from Program (VAnviteLli pEr la RicErca: VALERE) 2019 financed by the University of Campania "L. Vanvitelli".

## Author Contributions

**Conceptualization:** Antonio de Candia, Alessandro Sarracino, Lucilla de Arcangelis.

**Data curation:** Antonio de Candia, Ilenia Apicella.

**Formal analysis:** Antonio de Candia, Alessandro Sarracino, Ilenia Apicella, Lucilla de Arcangelis.

**Funding acquisition:** Antonio de Candia, Alessandro Sarracino, Lucilla de Arcangelis.

**Investigation:** Antonio de Candia, Alessandro Sarracino, Lucilla de Arcangelis.

**Methodology:** Antonio de Candia, Alessandro Sarracino, Lucilla de Arcangelis.

**Project administration:** Lucilla de Arcangelis.

**Supervision:** Lucilla de Arcangelis.

**Validation:** Alessandro Sarracino.

**Writing – original draft:** Antonio de Candia, Alessandro Sarracino, Lucilla de Arcangelis.

**Writing – review & editing:** Antonio de Candia, Alessandro Sarracino, Lucilla de Arcangelis.

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
