## [Decision Letter · Decision Letter 0]

14 May 2021

Dear Dr. Sarracino,

Thank you very much for submitting your manuscript "Critical behaviour of the stochastic Wilson-Cowan model" for consideration at PLOS Computational Biology.

As with all papers reviewed by the journal, your manuscript was reviewed by members of the editorial board and by several independent reviewers. In light of the reviews (below this email), we would like to invite the resubmission of a significantly-revised version that takes into account the reviewers' comments.

Of particular importance are the points raised by reviewer 2 about linking the results to other computational models of large-scale neural activity which address brain stability and criticality. A comprehensive review of the relevant literature, which covers different approaches and perspectives within the critical brain dynamics community, would be an important added value to the manuscript.

Assessing whether the stochastic WCM conforms with the “crackling noise” scaling relation, as suggested by reviewer 3, would also strengthen the results. 

We recognize that fully addressing the reviewers' comments may represent a significant challenge, and we leave it up to the authors if they prefer to submit the work elsewhere.

We cannot make any decision about publication until we have seen the revised manuscript and your response to the reviewers' comments. Your revised manuscript is also likely to be sent to reviewers for further evaluation.

Sincerely,

Oren Shriki, PhD

Guest Editor

PLOS Computational Biology

Lyle Graham

Deputy Editor

PLOS Computational Biology

Reviewer's Responses to Questions

**Comments to the Authors:**

Reviewer #1: See attachment

Reviewer #2: see attachment

Reviewer #3: de Candia et al present a numerical and analytical study of the stochastic Wilson-Cowan model as a potential explanation for critical dynamics observed in experimental observations of diverse neural systems. Interestingly, the same model was used (about a decade ago also in PCB by Benayoun et al) to make approximately the opposite claim - that the stochastic Wilson-Cowan model creates something that looks like critical dynamics without actually having the right underlying mechanisms to really, truly be critical dynamics. de Candi et al show that multiple phenomena predicted to occur for real and true critical dynamics do, in fact, occur for the stochastic Wilson-Cowan model (WCM). Thus, de Candia et al draw the conclusion that the WCM does have a real, bona fide critical point. A key advance that allowed them to reach their conclusions (in contrast with Benayoun et al) was that they used the Gillespie algorithm to numerically simulate very large network sizes, thus revealing some of the predictions of critical phenomena.

In my view, this paper is interesting and on a timely topic with a long-lasting debate surrounding it. However, there are multiple ways the paper could be improved. Below I outline first the bigger concerns and comments, followed by smaller comments.

Big Comment 1: One of the more commonly used aspects of critical phenomena in modern system neuroscience is the “crackling noise” scaling relation, which relates the average avalanche duration to the avalanche size according to a third scaling exponent. This has been used by many prominent studies to make claims about criticality: Friedman et al PRL 2012; Shew et al Nature Phys 2015; Ma et al Neuron 2019 and others. Does the stochastic WCM also conform to this scaling law? In my view, adding this to the current paper would make it substantially stronger and more related to the state of the art in this field. Moreover, this would be a much more interesting addition to the paper than the current diversion about different ways of defining avalanches (time bins vs. time series thresholding), which could be moved to methods or supplementary material.

Big Comment 2: Building on Big Comment 1, in my view the large digression (Figs 5-7) about how to define avalanches is not a very interesting addition to the paper. This disrupts the flow of the paper and it is very easy for a reader to lose interest. Perhaps it would be better to combine Figs 5-7 into one figure. Maybe even move it to the methods section or supplementary materials.

Big Comment 3: The section titled “Requirements to assess critical behavior” is difficult to follow for someone who is not an expert on statistical physics of critical phenomena. As a step towards improving understandability, I suggest that the authors stick with a more explicit description of the Ising model and how it relates to assessing critical behavior. This would allow the authors to state more clearly what an order parameter is (magnetization), and what a control parameter is (temperature), and what an “external field” is, etc. By making it more specific, less general, the reader has a better chance of following along.

Medium Comment 1: Considering that omega_0 is your control parameter, it would be helpful to provide more interpretation of the meaning of omega_0. I guess it represents a sort of e/i imbalance? It would also be helpful if the authors remind the reader of this meaning in the figure captions (maybe even in the axes labels of the plots).

Medium comment 2: When discussing and introducing the different alternatives for defining avalanches (time bins vs. time series threshold) it would be appropriate to cite some of the original experimental uses of these methods. For instance, Beggs & Plenz J Neurosci 2003 were among the first (maybe the first?) to use the time bin method. And Gautam et al PLos Comp Biol 2015 were among the first to use the time series thresholding methods.

Small comments

Line 11: what is a “front of independent neurons”. Consider rewording

Line 215: Should cite some relevant experiments

Fig 3 caption: Either get rid of the panel labels A-F or mention them in the caption

Fig 3 caption: typo: single

**Have the authors made all data and (if applicable) computational code underlying the findings in their manuscript fully available?**

Reviewer #1: **No: **In the version sent to me there is no references to available code

Reviewer #2: Yes

Reviewer #3: **No: **I did not see a proposed plan to make the code publicly available

PLOS authors have the option to publish the peer review history of their article (what does this mean?). If published, this will include your full peer review and any attached files.

Reviewer #1: **Yes: **Osame Kinouchi

Reviewer #2: No

Reviewer #3: No
---

## [Decision Letter · Decision Letter 1]

31 Jul 2021

Dear Dr. Sarracino,

We are pleased to inform you that your manuscript 'Critical behaviour of the stochastic Wilson-Cowan model' has been provisionally accepted for publication in PLOS Computational Biology.

Best regards,

Oren Shriki, PhD

Guest Editor

PLOS Computational Biology

Lyle Graham

Deputy Editor

PLOS Computational Biology

Reviewer's Responses to Questions

**Comments to the Authors:**

Reviewer #1: The authors provided answers to all my concerns.

Reviewer #3: The authors have responded constructively and completely to all of my previous concerns. I have no further concerns and recommend publication now.

**Have the authors made all data and (if applicable) computational code underlying the findings in their manuscript fully available?**

Reviewer #1: Yes

Reviewer #3: None

PLOS authors have the option to publish the peer review history of their article (what does this mean?). If published, this will include your full peer review and any attached files.

Reviewer #1: **Yes: **Osame Kinouchi

Reviewer #3: **Yes: **Woodrow L Shew

---

## [Editor Report · Acceptance letter]

13 Aug 2021

PCOMPBIOL-D-21-00435R1 

Critical behaviour of the stochastic Wilson-Cowan model

Dear Dr Sarracino,

I am pleased to inform you that your manuscript has been formally accepted for publication in PLOS Computational Biology. Your manuscript is now with our production department and you will be notified of the publication date in due course.

With kind regards,

Andrea Szabo
